# Preparation of Ag Doped Keratin/PA6 Nanofiber Membrane with Enhanced Air Filtration and Antimicrobial Properties

**DOI:** 10.3390/polym11091511

**Published:** 2019-09-16

**Authors:** Baolei Shen, Dongyu Zhang, Yujuan Wei, Zihua Zhao, Xiaofei Ma, Xiaodan Zhao, Shuo Wang, Wenxiu Yang

**Affiliations:** 1College of Textile and Garments, Hebei University of Science and Technology, Shijiazhuang 050018, China; shenbaolie@163.com (B.S.); weiyujuan66@sina.com (Y.W.); mxiaof1007@163.com (X.M.); 2School of Environmental Science and Engineering, Hebei University of Science and Technology, Shijiazhuang 050018, China; dongyuz01@163.com; 3College of Bioscience and Bioengineering, Hebei University of Science and Technology, Shijiazhuang 050018, China; zzh1398921@163.com (Z.Z.); zxd199404@163.com (X.Z.)

**Keywords:** coarse wool, keratin, electrospinning, antimicrobial, filtration

## Abstract

Coarse wool is a kind of goat wool that is difficult to further process in the textile industry due to its large diameter, dispersion, better strength, and less bending. Therefore, coarse wool is often discarded as waste or made into low-cost products. In this work, keratin was extracted from coarse wool by a high-efficiency method, and then, an Ag-doped keratin/PA6 composite nanofiber membrane with enhanced filtration and antibacterial performance was prepared using HCOOH as solvent and reductant. HAADF-STEM (high-angle annular dark field-scanning transmission electron microscopy) shows that AgNPs are uniformly distributed in keratin/PA6 (30/70) nanofibers. TGA (Thermogravimetric Analysis) and DSC (Differential Scanning Calorimetry) were employed to investigate the thermal stability of composite membranes with different keratin and AgNP contents. The present keratin as a dopant with polyamide-6 (PA6) was found not only to improve air filtration efficiency but also to enhance water–vapour transmission (WVT). The addition of the Ag nanoparticles (AgNPs) gave a strong antibacterial activity to the composite membrane against *Staphylococcus aureus* (99.62%) and *Escherichia coli* (99.10%). Bacterial filtration efficiency (BFE) of the composite membrane against *S. aureus* and *E. coli* were up to 96.8% and 95.6%, respectively. All of the results suggested a great potential for coarse wool extraction and application in the air filtration field.

## 1. Introduction

China is one of the largest cashmere producers of the world; cashmere as a kind of textile material is widely used in various garment products. As far as China is concerned, more than 200,000 t of cashmere are yielded every year. Raw goat cashmere contains cashmere, impurities, and coarse wool. Coarse wool is a kind of scrap in the raw cashmere-carding process. Coarse wool is rigid, has less bending, and has poor spinnability, which has limited its application in the textile industry, so it is often discarded as waste or processed into low-cost products [1], which not only wastes resources but also increases the cost of the factory. Therefore, expanding the utilization of coarse wool can bring great economic benefits to enterprises.

Keratin is the main component of the outer epidermal scales and fibrils of wool. Because of its naturally hydrophilic, biocompatible, and biodegradable characteristics, the development of wool keratin has attracted wide attention from scholars all over the world [2,3,4,5]. Cystine was found in keratin to be able to form disulfide-bond inter- and intra-chains and three-dimensional network structures that endowed the wool fiber with superior structural stability, fair softness, and elasticity due to cross linking [6,7], which made it difficult to extract keratin from wool. So far, various measures have been taken to extract keratin from waste wool, such as chemical methods, physicochemical methods, and biological enzyme methods, etc. [8,9,10]. The physicochemical and enzymatic hydrolysis methods need higher cost, have long production cycles, and have low extraction rates, which have limited their development. The widely used means is chemical methods, including oxidation and reduction, which, when attributed to the oxidizing and reducing agent, can destroy the disulfide bonds of the wool. Wool keratin with excellent biocompatibility, hydrophilicity, environmental friendliness, and sustainability could be used in various field, such as maquillage, medication, and filter materials [2,5].

Particulate matter (PM) of airborne pollutants poses a severe threat to human respiratory health. PM2.5 (particles having an average size in the adjacent area of 2.5 μm) is the widespread pollutant easily inhaled into lungs and even diffused to other organs, increasing cardiac and respiratory morbidity [11,12]. Nanofibers with high filtration efficiency and low resistance show great application potentials in the field of air filtration. Theoretically, the slip-flow effect, which is beneficial to improving the performance of filtration, can be seen when the diameter of nanofibers is less than 500 nm [13]. Moreover, a high specific and surface area of nanofibers is also helpful to improving filtering efficiency [14]. 

Electrospinning is one of the most effective methods with low cost and versatility to prepare nanofibers of controllable fiber diameter and pore size. Electrospun nanofibers are also widely applied in many fields, such as filtration [15], wound repair [16,17], biological scaffold [18], and so on. However, the poor viscosity and brittle properties of wool keratin make it difficult to prepare nanofibers alone [19,20]. The most effective way to broaden the application of keratin is blending keratin with other suitable polymers [21,22,23]. Aluigi produced randomly oriented nanofiber mats by electrospinning a keratin/PA6 blending solution and assessed their performance as adsorbents of heavy metal ions [1,24]. Keratin extracted from human hair was composited with PCL (poly (ε-caprolactone)), which was an aliphatic linear polyester with a slow degradation rate and hydrophobicity. TFE (Trifluoroethanol) as nanofiber mats could support the growth and proliferation of 3T3 cells (a mouse fibroblast cell line) [25,26]. 

During the preparation and use of filter materials, it is inevitable to be exposed to bacteria, which will seriously affect their shelf life. As a kind of broad-spectrum antibacterial agent, silver nanoparticles have been applied in various antibacterial products [27,28]. The transformation of Ag nanoparticles into composite nanofibers would be an interesting strategy to product nanofiber mats with strong antibacterial properties [29]. 

In this article, a new chemical method for extracting keratin from coarse wool by a Na_2_S_2_O_3_ mixed solution has been developed and the extraction rate of keratin from coarse wool with a molecular mass less than 31 kDa was 75.3%. Then, the composite nanofiber membrane (Ag-keratin/PA6) with strong antibacterial and high filtration performances was prepared using formic acid as solvent and reductant. The different diameter distributions of the nanofiber mats prepared by different blending rates of Ag-keratin/PA6 were investigated. AgNO_3_ was reduced to AgNPs in a composite solution by formic acid. A flowchart describing the preparation process of an Ag-keratin/PA6 composite membrane is illustrated in Figure 1. Moreover, the study of quality factor (QF, as shown in Equation (1)), moisture permeability, and antibacterial and bacterial filtration efficiency (BFE) of the composite nanofiber mats suggested their application potential in the air filtration field.

## 2. Materials and Methods

### 2.1. Materials

Coarse wool was obtained from Hebei Yuteng Cashmere Products Co., Ltd. (Qinghe, China). PA6 (1013B) was purchased from UBE Co. Ltd. (Tokyo, Japan). Tris, 10% sodium dodecyl sulfate (SDS), Coomassie brilliant blue R-250, and ammonium persulfate were obtained from Sigma; PAGE pre-solution and protein sample buffer and marker (14.4–97.4 kD) were obtained from Solarbio. *Staphylococcus aureus* (ATCC 6538) and *Escherichia coli* (ATCC 11229) were obtained from SMIC Qiheng Scientific Instrument Co., Ltd. (Suzhou, China). All the other reagents were used directly without further purification.

### 2.2. Keratin Extraction and Blend Membranes Preparation

#### 2.2.1. Preparation of the Ag-Keratin/PA6 Composite Solution

Keratin was extracted from the coarse wool by the reduction method. Lanolin was removed by the ultrasonic process in a sodium hydroxide solution (5 g/L) for 30 min, and then, the coarse wool fibers were washed with deionized water and dried at 70 °C. The keratin was obtained by putting a certain amount of fibers (1 g) into a mixed solution containing 7.5 g urea, 0.5 g sodium thiosulfate, 0.1 g SDS, and 25 mL water and by stirring for 90 min at 120 °C. After that, the solution was dialyzed for two days [30] in a dialysis bag with the molecular weight 3500 Da and dried at 50 °C.

The Ag-keratin/PA6 composite solution was prepared by adding the keratin powder and AgNO_3_ to a formic acid solution of PA6 and stirred at room temperature for 5 h to obtain a 28.5 wt% (keratin and PA6) composite solution. In the process, AgNO_3_ could be reduced to nano-silver. 

#### 2.2.2. Preparation of Ag-Keratin/PA6 Composite Membrane

An electrospinning apparatus was employed to prepare composite nanofiber mats with keratin contents of 0%, 30%, 50%, and 70%, which were denoted as Ker.0%/PA6, Ker.30%/PA6′ (this keratin was extracted by a reduction method using NaHSO_3_ as the reductant [9]), Ker.30%/PA6, Ker.50%/PA6, and Ker.70%/PA6, respectively. The composite solution was put into a 15-mL syringe with a 20 G flat-tip needle and a metal roll as the spinneret. In addition, electrospinning was performed at 25 °C and humidity of 20% for 10 h under the conditions of feeding rate 0.1 mL/h and 20 kV with receiving distance 25 cm. This experiment accorded with the basic law of electrospinning, which is not discussed in this paper.

AgNPs wrapped in the composite membrane with antimicrobial function was obtained by dissolving AgNO_3_ in a keratin/PA6 mixed solution and was reduced by their solvent [31]. Therefore, the Ag-doped keratin/PA6 electrospun solution was obtained by the abovementioned operation (Ag/(keratin + PA6) = 0.01 and 0.1). The Ag-doped keratin/PA6 nanofibers were prepared [31] also under the conditions of feeding rate 0.1 mL/h and 20 kV with receiving distance 25 cm. In addition, the Ker.30%/PA6 composite membrane with AgNP contents of 1% and 10% were denoted as Ag.1%-Ker.30%/PA6 and Ag.10%-Ker.30%/PA6, respectively.

### 2.3. Characteristic

#### 2.3.1. Determination of Molecular Weight of Keratin

An SDS-PAGE gel-electrophoretic apparatus was used to determine the relative molecular mass of keratin [30]. The test was carried out in a container with concentrated glue (5 wt.%) as the upper layer and separation gel (15 wt.%) as the bottom layer at voltages of 80 V and 140 V. After that, gels were stained with Coomassie brilliant blue R-250 and detained.

#### 2.3.2. Characterization of the Fibers

The micromorphology of nanofibers was observed by field-emission scanning electron microscope (FE-SEM, S-4800-1, Hitachi, Tokyo, Japan) at an accelerating voltage of 5 kV. The samples were sputter-coated with gold. The average diameter of nanofibers was measured from at least 100 nanofibers per sample by image-pro plus software [32]. A transmission electron microscope (TEM, JEM 2010, Tokyo, Japan) was used to confirm the distribution of AgNPs in the AgNP-doped composite nanofibers.

#### 2.3.3. Filtration Performance Testing

Two DustTrak aerosol monitors (Model 8530, TSI, MN, USA) were performed to measure the concentration of PM2.5 on both sides of the composite membranes (downstream and upstream) in a polluted-air environment in Shijiazhuang, and air filtration efficiency was calculated based on the difference of concentration. At the same time, the pressure drop of gas was measured by a pressure sensor. The QF was used to evaluate the overall filtration performance of the composite membranes [29].
(1)$$QF=\frac{-In(1-E)}{\Delta P}$$
where *E* is the filtration efficiency and Δ*p* is the pressure drop. This equation is usually used to evaluate the overall filtration performance of a filter.

#### 2.3.4. Pore Size Test

Pore size of the composite membranes was carried using a capillary flow porometer (CFP-1100AX, New York, NY, USA). A circular sample (diameter of two centimeters) was placed into the instrument for testing. Then, average pore size and pore size distribution of the samples could be obtained from distribution data diagrams of the composite membranes.

#### 2.3.5. Water–Vapour Transmission (WVT) of the Composite Films

The water–vapour transmission test of the nanostructured mats was performed by using ASTM E96E96M-2005 Standard Test Methods, and computer-type fabric moisture permeability testing apparatus (YG601-I/II) was performed in this procedure. A circular specimen (diameter of 7 cm) and suitable distilled water filled the test dish. Significantly, in order to reduce the risk of water touching the specimen when the dish is handled, it is necessary to leave a little space between the water and samples; otherwise, the results would not be convincing. The test dishes should be weighed before and after testing, and the masses were *m*_1_ and *m*_2_, respectively. The test was performed at 23 ± 2 °C, and relative humidity was 50 ± 2% for 1 h. WVT is used to estimate the moisture permeability of the blending nanofiber mats, which is defined as follows:(2)$$WVT({g/m}^{2}\cdot d)=\frac{\Delta m-\Delta m\prime }{At}$$
where *WVT* (g/(m^2^·h)) is the water–vapour transmission rate, △*m* is the difference between the two weights (△*m* = *m*_1_ − *m*_2_) of the same test dish in grams (g), △*m*′ is the difference between the two weights of the same test dish of a blank sample, *A* is the effective test area (where *A* is 0.00283 m^2^), and *t* is time (h). This equation is usually used to evaluate the moisture permeability of materials.

#### 2.3.6. TGA Analysis

Simultaneous thermogravimetric analysis (DTG-60H, Shimadzu, Japan) was used to evaluate the thermogravimetric of composite films; 2 mg of the film was put into an alumina crucible for this analysis. The test was carried out at 30–800 °C with a heating rate of 10 °C/min and with nitrogen flushed at 70 mL/min. Derivative thermogravimetry (DTG) was performed to record the temperature of maximum mass-loss rates [33].

#### 2.3.7. Differential Scanning Calorimetry (DSC) Analysis

The thermal stability of Ag-keratin/PA6 composite membranes were analyzed by differential scanning calorimetry (Netzsch DSC 214). The test was performed from 20–300 °C, with a heating rate of 20 °C/min and by flushing nitrogen at 100 mL/min.

#### 2.3.8. Antibacterial Properties of AgNP-Doped Composite Films

Antibacterial properties of AgNP-doped composite films were investigated against *E. coli* (ATCC 11229) and *S. aureus* (ATCC 6538) using ISO 20645 Standard Test Methods. Suspensions of *E. coli* and *S. aureus* (100 μL each) with 10^8^ CFU/mL were uniformly dispersed on the culture dish. Then, samples (not sterilized) of AgNP-doped composite films and primary keratin/PA6 films were pasted on agar plates and incubated at 37 °C for 48 h. An optical microscope was used to judge the antibacterial status of the composite film easily (interior and edge) [29]. In order to check the antibacterial activity of the composite membrane, another antibacterial test method was used. In the process, the nutrient broths of *S. aureus* and *E. coli* were diluted to the required concentrations by PBS (phosphate buffer saline) buffer (3 × 10^5^ CFU/mL). Then, 75 mL of diluted bacteria suspension with 0.75-g samples were incubated at 24 °C with gentle shaking (100 rpm) in a shaking table for 24 h. Bacterial colony count was used to evaluate the bacterial inhibition rate (BI) of AgNP-doped composite films by the following equation [34]:(3)$$BI=\frac{B-A}{B}\cdot 100\%$$
where *A* is the number of bacterial colonies with Ker.30%/PA6 and *B* is the number of bacterial colonies with Ag-Ker.30%/PA6.

#### 2.3.9. Bacterial Filtration Efficiency (BFE) Analysis

Bacterial filtration efficiency (BFE) tests were carried out by employing gram-positive *S. aureus* and gram-negative *E. coli* using ASTM F2101-14 Standard Test Method. In this process, the number of *S. aureus* and *E. coli* in liquid culture medium was determined by the colony count method and the nutrient broth was diluted to the required concentration by a PBS buffer [35]. Then, the liquid was used to prepare microbial aerosol by the spray suspension method and the challenge suspension was delivered to the nebulizer in a Handersen pipeline for 1 min. Then, the air pressure and cascade impactor ran for 2 min. The sample port with the nanofiber mats and non-mats were counted with an Anderson 6 level microbial aerosol sampler at a flow rate of 28.3 L/m. Then, the specimen plates were incubated at 37 °C and the colony was counted after 48 h. In addition, the count correction table of the Anderson sampler was used to correct the number of colony counts. The filtration efficiency percentages were calculated using the following equation:(4)$$BFE\%=\frac{C-T}{C}$$
where *C* is the average plate count for test controls and *T* is the plate count total of the test sample.

## 3. Results and Discussion

### 3.1. Determination of the Molecular Weight of Keratin

Coarse wool is a kind of scrap in the raw cashmere-carding process with a diameter of about 60 μm, which is much larger than commercial wool (diameter of about 13 μm; Appendix A). In addition, coarse wool contains more dense internal structures and higher crystallinity, so it has greater decomposition difficulty than commercial wool. Herein, a stronger reducing agent, Na_2_S_2_O_3_, compared with NaHSO_3_ was engaged to extract keratin from coarse wool (as shown in Section 2.2.1). The extraction rates of keratin from coarse wool were 75.3% (the Na_2_S_2_O_3_ method, named M1) and 23.2% (the NaHSO_3_ method, named M2).

Gel electrophoresis analysis was performed to identify keratin with the two samples (M1 and M2; Appendix A). Both specimens showed high intensity bands below 21 kDa, but the latter, M2, had different bands at around 31 to 97.4 kDa. However, there are no sharp stripes here for the reason that reductants can destroy various chemical bonds and then a series of mixtures of hydrolysate can be obtained [30,36].

### 3.2. Structural Characterization of the Nanofibers

The morphology of Ag-Keratin/PA6 composite membranes with different keratin contents from 0% to 70% are shown in Figure 2. It indicates that all samples have a uniform structure with a random orientation and wide, large fiber diameters.

The Ag-keratin/PA6 composite solutions were electrospun within 24 h to avoid the separation of various polymers due to their insolubility. Sure enough, the viscosity of a mixed solution decreases with the increase in keratin content while the conductivity of a mixed solution increases with the decrease in keratin content. Doped AgNPs could also significantly improve conductivity (Table 1). It has been proven that the diameter of nanofibers decreases with the increase of conductivity and decrease of viscosity [32,37]. In this system, both keratin and doped AgNPs could change the viscosity and conductivity of the composite solution and can affect the diameter of nanofibers. On the one hand, the increase of conductivity brings about more charges on the jet process and will be subjected to greater electric-field forces in an electric field to make nanofibers thinner. On the other hand, the lower viscosity of the solution gives a lower viscoelastic force, which is beneficial in forming electrospun jets, and results in fiber formation. The diameters of composite membranes with different keratin contents decrease from 293 nm to 155 nm due to the changes in conductivity and viscosity of the solution. These results are similar to those reported for keratin/PEO (polyethylene oxide) [9] and keratin/PVA (polyvinyl alcohol) [38]. In addition, the diameters of Ker.30%/PA6 decrease from 192 nm to 168 nm with the addition of AgNPs, which could be explained by the conductivity of the solution (Table 1). HAADF-STEM and the corresponding EDS (energy dispersive spectrometer) mapping (Figure 2i) show that the Ag and S (only in keratin) elements are uniformly distributed in the electrospun nanofibers.

### 3.3. Filtration Performance of the Nanofiber Film

WVT presents the water–vapour transmission capacity of the composite membranes. The higher the WVT, the better the water–vapour transmission performance is. The QF is used to evaluate the comprehensive filtration performance of the composite membrane. The higher the QF value, the better the filtration performance is. Figure 3 shows the filtration performance of the nanofiber films against PM2.5. The mean pore sizes decrease gradually from 1.398 μm to 0.633 μm with keratin content increase. It could be explained by two aspects: on the one hand, the existence of keratin breaks the force between and inside the PA6 molecules and then the solution viscosity decreases (as shown in Table 1), which results in a smaller fiber diameter. On the other hand, the addition of the AgNPs increases the conductivity of the composite solution and enhances the electric force, which also leads to a smaller fiber diameter. WVT values of different nanofiber mats decrease from 166.2 g/m^2^·d to 146.2 g/m^2^·d with keratin increase. When the content of keratin is 30% (166.2 g/m^2^·d), the WVT value is higher than that of Ker.0%/PA6 nanofiber mats (151.2 g/m^2^·d). The possible reason is that the addition of the keratin increases the hydrophilicity of the composite membrane. However, the WVT value decreases gradually with the increase in keratin content for the continuously decreasing pore size diameter (Figure 3b; the specific aperture distribution is shown in Appendix A and Appendix A). QF values also change with different keratin contents. When the content of keratin is 30% (0.044), the QF value is better than that of Ker.0%/PA6 nanofiber membrane (0.021). It could be ascribed to the higher filtration efficiency and lower resistance. Moreover, the addition of the keratin imparts the composites with excellent flexibility and permeability and causes lower filtration resistance. When the content of keratin continues to increase, the QF value worsens. It could be attributed to the smaller pore size, which increases filtration resistance and plays a major role in the filtration property of the composite membranes (the filtration efficiency and filtration resistance are shown in Appendix A). Notably, the same regular pattern is also applicable to composite membranes with large molecular weight for keratin extracted using the M2 method. However, the QF value of Ker.30%/PA6′ is not as good as Ker.30%/PA6 mats due to keratin (M2) wrapping being unable to provide enough hydrophilic groups to counteract the effects of pore size reduction compared with keratin extracted using the M1 method.

### 3.4. TG Analysis

The thermal stability is used to study the effect of the structure and properties of PA6 nanofiber membranes with keratin and doped AgNPs, and it is also beneficial in adjusting the parameters in the electrospinning process.

In Figure 4, it is possible to see the thermal behavior difference between Ker.0%/PA6 (pure PA6) and the blend mats. All samples display two distinct weight-loss regions in the 30–800 °C range. The initial weight loss from 100% to 90% results from water evaporation (Figure 4a) [31]. The second weight loss is caused by the degradation of nanofibers. Ker.0%/PA6 shows a higher thermal stability throughout the whole test, with a higher initial degradation temperature (380 °C) and a peak degradation temperature at 466 °C. As the content of keratin increases (from 30% to 70%), the initial degradation temperature (from 371 °C to 262 °C) and endothermic peak (from 432 °C to 408 °C) of the composite system decreases, which indicates that doped keratin could make the thermal stability of the composites decrease. However, the addition of keratin could slightly improve the residual weight from 0.92% to 13.27% (Figure 4a), which is attributed to the lower degradation temperature and larger residual mass of keratin compared to that of Ker.30%/PA6 [34,36,39]. In addition, doped AgNPs and keratin (NaHSO_3_ method with a large molecular weight, M1) could also change the thermal behavior of the complex film. Especially, AgNPs could significantly improve the residual weight of composites nanofibers. There is about 22.6% residual mass in Ag.1%-Ker.30%/PA6 and 30.1% in Ag.10%-Ker.30%/PA6 nanofibers at 800 °C (Appendix A), which is more than that of normal composite nanofibers. Otherwise, the significant increase of the residual weight also confirms the presence of AgNPs in nanofibers.

### 3.5. DSC Analysis

The melting endotherms of PA6, keratin/PA6, and Ag-keratin/PA6 nanofiber films were investigated by means of DSC performed under nitrogen atmosphere. Figure 5 shows the melting endotherm of PA6, keratin/PA6′, keratin/PA6, and Ag-keratin/PA6 nanofiber films tested by DSC. Ker.0%/PA6 shows a doublet endotherm, where the lower temperature peak (207 °C) of the γ crystalline structure is associated with the main peak at 218 °C of the α-form [40]. In the composite nanofiber films, the γ-form melting peak of PA6 increases with low content of keratin and disappears when the keratin content reaches a higher degree. Ker.30%/PA6 shows a higher melting peak (219 °C) because of directional stretching of the composite nanofibers during electrostatic spinning, which could facilitate the formation of more crystal structures [4]. As for composite nanofibers, when the keratin contents are more than 30%, the melting peaks of the composite films decrease gradually from 219 °C to 209 °C. The addition of AgNPs and keratin (NaHSO_3_ method with a large molecular weight, M2) could also influence the hydrogen bonding strength between keratin and PA6, especially when the content of AgNPs reach 10% and the melting peak of PA6 drops to 196 °C (the peak of melting endotherms of the nanofibers mats are shown in Appendix A). During the rapid evaporation of the solvent in the preparation of nanocomposites, the existence of keratin and AgNPs interferes with the formation of PA6 crystal structures.

### 3.6. Antibacterial AgNP-Doped Composite Nanofibers

As we all know, the air we breathe contains a lot of bacteria (including *E. coli* and *S. aureus)* and most of them can enter the body as people breathe. These microbes may cause the spread of numerous diseases [29]. Therefore, it is necessary to prepare air filtration membranes with antibacterial properties in order to reduce the inhalation of bacteria. AgNPs wrapped in keratin/PA6 nanofibers could enhance their antibacterial performance.

The antibacterial of AgNP-doped composite nanofibers was studied by surveying the bacterial growth-inhibition halos and bactericidal kinetics against *E. coli* and *S. aureus.* As shown in Figure 6a,b,d,e. Bacterium grew well on Agar plates with different matrices for 24 h. The Ker.30%/PA6 and Ag.1%-Ker.30%/PA6 nanofibers mats showed no obvious antibacterial activity, and bacteria could grow normally around the composite membranes. However, a clear bacterial inhibition zone around the Ag.10%-Ker.30%/PA6 specimen could be seen. Meanwhile, mould colonies emerged around the Ker.30%/PA6 composite membranes (Figure 6a,d). As shown in Table 2, the bacterial inhibition rates against *S. aureus* (*E. coli*) are 0% (0%), 20.65% (19.95%), and 99.62% (99.10%) for the Ker.30%/PA6, Ag.1%-Ker.30%/PA6, and Ag10%-Ker30%/PA6 fibrous membranes, respectively. Clearly, the addition of AgNPs has a great influence on the antibacterial activity of the composite membrane and the antibacterial effect increases significantly with the increase of silver contents. On the other hand, the bacterial filtration efficiency (BFE) of Ag.10%-Ker.30%/PA6 was studied by investigating the number of bacteria at the filtration material and unfiltration material sample ports using ASTM F2101-14 Standard Test Method. The average filtration efficiencies of *S. aureus* and *E. coli* were 96.8% and 95.6%, respectively. These results indicate that Ag.10%-Ker.30%/PA6 nanofiber membranes could be used to prepare bioprotective filter materials. Meanwhile, we simulated the working environment of air filtration membranes and tested the bacterial inhibition rate of the composite membrane (Ag.10%-Ker.30%/PA6) before and after continuous operation for 120 h under the air filtration condition (PM_2.5_ > 1500 μg/m^3^) to evaluate the durable antibacterial activity of Ag.10%-Ker.30%/PA6 nanofiber membranes [41]. The nanofiber membranes of Ag.10%-Ker.30%/PA6 maintained a high level of antibacterial effectiveness, and the bacterial inhibition rate against *S. aureus* (*E. coli*) decreased from 99.62% (99.10%) to 96.7% (95.4%). It could be attributed to the antibacterial properties of doping silver ions decaying very slowly with the filter materials worn out [35].

## 4. Conclusions

In summary, an efficient method for extracting keratin from coarse wool by a Na_2_S_2_O_3_ mixed solution has been developed and a composite air filtration membrane (Ag.10%–Ker.30%/PA6) with an average diameter of 168 nm and an excellent water–vapour transmission rate (WVT) and air filtration performance was prepared using HCOOH as solvent and reductant. The addition of keratin was found not only to improve the air filtration efficiency but also to enhance water–vapour transmission (WVT). The addition of the Ag nanoparticles (AgNPs) gave the composite membrane strong antibacterial activity against *S. aureus* (99.62%) and *E. coli* (99.10%). Meanwhile, bacterial filtration efficiency (BFE) of the composite membranes against *S. aureus* and *E. coli* were up to 96.8% and 95.6%, respectively. These results not only suggest a great potential of the composite nanofiber membrane as comfortable and personal bioprotective air filters but also expand the utilization of coarse wool and could bring great economic benefits to enterprises.

## Figures and Tables

**Figure 1 polymers-11-01511-f001:**
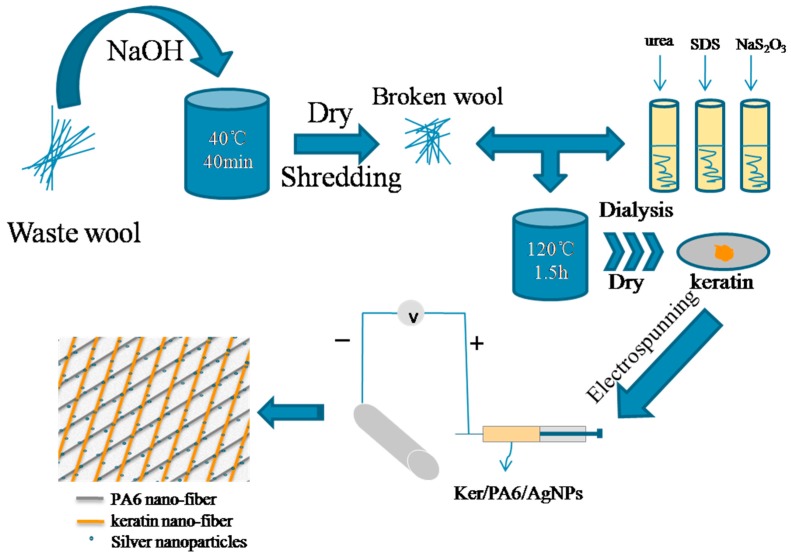
Schematic diagram of the wool keratin extraction and preparation of an electrospun keratin/PA6 air filtration membrane with antimicrobial function.

**Figure 2 polymers-11-01511-f002:**
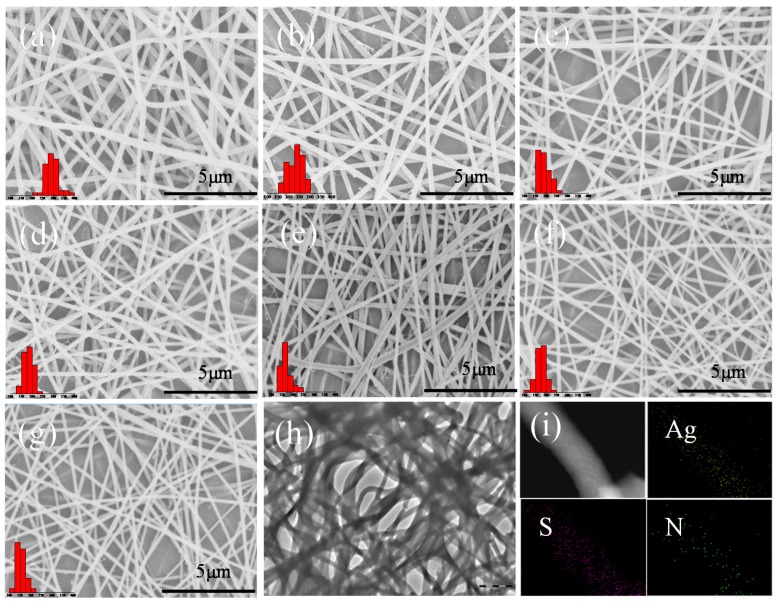
SEM images of fibrous membranes with different concentrations of polymer mats: (**a**) Ker.0%/PA6, (**b**) Ker.30%/PA6′, (**c**) Ker.30%/PA6, (**d**) Ag.1%-Ker.30%/PA6, (**e**) Ag.10%-Ker.30/PA6, (**f**) Ker.50%/PA6, and (**g**) Ker.70%/PA6. The histograms show the distribution of fiber diameters corresponding to their SEM image. (**h**) TEM images, (**i**) HAADF-STEM (high-angle annular dark field-scanning transmission electron microscopy), and the corresponding EDS (energy dispersive spectrometer) mapping of the Ag, S, and N elements in composite nanofibers.

**Figure 3 polymers-11-01511-f003:**
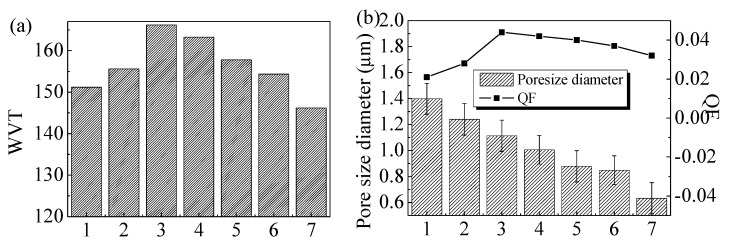
Schematic diagram of the quality factor (QF), pore size diameter, and water–vapour transmission (WVT) of composite membranes: (**a**) the WVT of composite membranes and (**b**) the comprehensive filtration performance and pore size of composite membranes. Columns 1–7 represent Ker.0%/PA6, Ker.30%/PA6′, Ker.30%/PA6, Ag.1%-Ker.30%/PA6, Ag.10%-Ker.30/PA6, Ker.50%/PA6, and Ker.70%/PA6, respectively.

**Figure 4 polymers-11-01511-f004:**
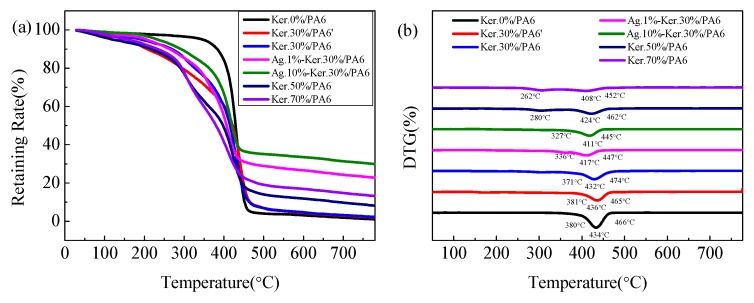
Thermogravimetric Analysis (TGA) (**a**) and Derivative Thermogravimetry (DTG) (**b**) of as-prepared nanofiber films.

**Figure 5 polymers-11-01511-f005:**
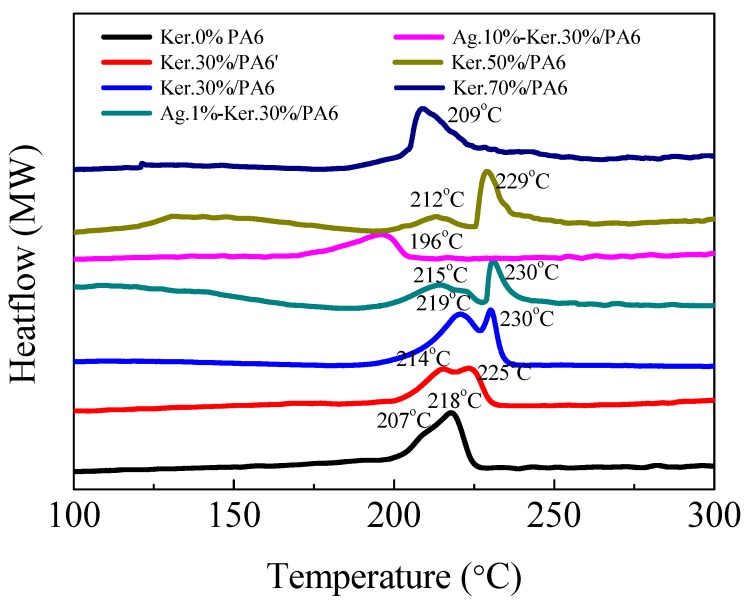
Differential Scanning Calorimetry (DSC) thermograms of different AgNPs and keratin content of PA6 composite nanofiber films.

**Figure 6 polymers-11-01511-f006:**
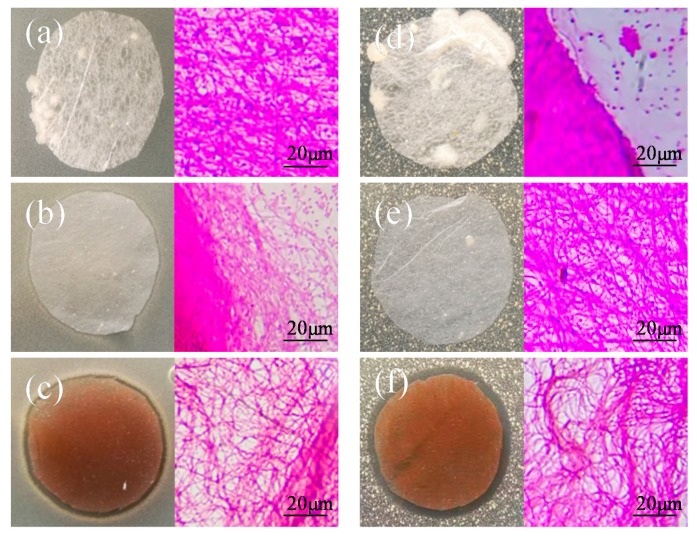
Schematic diagrams of the antibacterial activity of composite nanofibers: (**a**–**c**) *E. coli* and (**d**–**f**) *S. aureus*. Figure 6a,d, Figure 6b,e, and Figure 6c,f represent Ker.30%/PA6, Ag.1%-Ker.30%/PA6, and Ag.10%-Ker.30%/PA6, respectively. The right-hand images of Figure 6a–f correspond to optical microscopic images.

**Table 1 polymers-11-01511-t001:** Composite solution and nanofiber performance.

Samples	Diameter (nm)	Viscosity (Pa·S)	Conductivity (mS/cm)
Ker.0%/PA6	293 ± 70	2.48	3.18
Ker.30%/PA6′	229 ± 74	1.95	4.15
Ker.30%/PA6	192 ± 65	1.09	4.45
Ag.1%-Ker.30%/PA6	183 ± 40	0.94	4.67
Ag.10%-Ker.30%/PA6	168 ± 60	0.85	6.83
Ker.50%/PA6	174 ± 49	0.78	5.26
Ker.70%/PA6	155 ± 50	0.70	6.53

**Table 2 polymers-11-01511-t002:** Bacterial inhibition zone and bacterial inhibition rate of the AgNP-wrapped Ker.30%/PA6 blend nanofibers mats.

Samples	Bacterial Inhibition Zone (mm)		Bacterial Inhibition Rate (%)	
	*E. coli*	*S. aureus*	*E. coli*	*S. aureus*
Ker.30%/PA6	0	0	0	0
Ag.1%-Ker.30%/PA6	<1	0	19.95	20.65
Ag.10%-Ker.30%/PA6	>1	>1	99.10	99.62

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
