# Peer review of "Preparation of Ag Doped Keratin/PA6 Nanofiber Membrane with Enhanced Air Filtration and Antimicrobial Properties"

_polymers, 2019, doi:10.3390/polym11091511_

Round 1
Reviewer 1 Report
The Manuscript by Shen et al describes one step method for preparing Ag doped keratin/PA6 composite nanofiber membrane with high filtration and antibacterial activity against E. coli and S. aureus.
Major points:
Figure 2: I do not see any sharp bands at either 31kD or `14kD. There are only smears. Author should consider doing a western blot (antibody)or optimizing protein concentration to see the band.
This gel can also be moved to supplemental than main figure.
TGA analysis does not have any discussion. It seems like 30% is the which has best filtration properties. However, this result has not been discussed in the light of TGA result about thermal stability of polymer.
Figure 6. there are two temperature in the graph for each peak except ker30% and 70% . Is there specific reason for not adding the temperature in these two peaks. Again, there is absolutely no discussion that which condition is best in terms of endothermal properties. How that correlates to the result of filtration activity and thermal stability.
L246- The antibacterial activity is not strong. Did author also calculate the zone of clearance? If no, the authors need to give a zone of clearance or MIC. At least it is moderate for Staphylococcus. Although the optical microscopic data looks little convincing author should check that with Scanning electron microscopy to make sure it is indeed true.
How many areas was considered when the light microscopic experiment was performed? Is this result being representative for several areas looked at?
Please provide the identification of E. coli and S.aureus. what is the source? Are they bought from company then mention ATCC numbers?
Is there any evidence that these bacteria are prominently present in air? If yes, provide citation to explain why these are used for antibacterial properties test. In addition to this these two bacteria belongs two group called gram negative and positive which will help authors to say they have done it in broader range of bacteria.
The manuscript has a serious flaw of not citing work of others and comparing it with their result. Has no body ever tried to make this sort of polymer by using Ag or Keratin in the past? If no please cite their work as discussion with your current results.
None of the microscopic picture has scale bar. Please add.
L189 “the reason can be explained by two reasons” the sentence needs modification
The number keywords can be more.
I am not sure of email Ids “qq.com” please provide institutional email ID.
Extensive English editing is required i.e L38 “Consequently, the keratin is difficulty to dissolve”.
Font of fig ligand is different than main MS.
Author Response
Dear Reviewer
Thank you for reading our manuscript and reviewing it, which will help us improve it to a better scientific level. We revised our manuscript, and quit a lot of changes have taken place. So we have sent the revised manuscript, and a version containing all the changes to be visible. The answers for the questions are shown in attachment.

Reviewer 2 Report
The manuscript entitled ‘Preparation of Ag doped keratin/PA6 air filtration membrane with enhanced antimicrobial properties through one-step method’ by B. Shen et al. primarily discussed about the utilization of wool in developing antimicrobial air filter membrane. The main component of the wool, keratin was extracted from wool and then blended with PA6 (I assume PA6 is Polyamide, Nylon 6) to warrant a feasible membrane fabrication process by electrospinning method. Prior to electrospinning process, the keratin/PA6 blend solution was treated with silver nitrate solution to develop silver nanoparticles in the membrane for air filtration.
Firstly, the extraction of keratin from wool by sulphitolysis and blending with PA6 to facilitate electrospinning process is an established technology. Also, the use of silver nitrate solution to develop silver nanoparticles and to introduce antimicrobial functionality in the air filtration membrane are well known. So, there is no exciting science novelty in this work. The authors also did not highlighted the novelty of this research work. This is certainly a multi-step process and hence the claims for ‘one-step’ method is not valid.
The authors attempted a few characterization techniques such as gel-electrophoretic method to determine the molecular weight of the extracted keratin, SEM for morphological studies, TGA and DSC for thermal studies etc. However, it is important to know why determination of molecular weight is crucial such as whether there is any effect of molecular weight of keratin on the process optimization and/or membrane performance.
Similarly, it is not clear why the thermal stability and the degradation of composite nanofibers are important in air filtration application. Do they have any contribution on the key requirements such as process optimization and/or membrane performance?
One of the critical factor in the electrospinning technique is the process optimization, which is not discussed. The SEM images are too dull. How is it possible to accurately measure the fiber diameter in nm scale from the micron scale images? The presence of Ag nanoparticles in the nanofibers is also not evident from these images. Though the authors mentioned about solution viscosity, conductivity etc but they did not provide any experimental solution viscosity, conductivity data and did not try deep analysis.
Most importantly, the effectivity of the membrane against critical pathogenic species commonly found in air filters such as Pseudomonas aeruginosa and Legionella pneumophila are not provided. The methodology provided for antibacterial studies is not adequately robust to claim the long lasting strong antibacterial activity. For this application, dosing bacterial aerosol under controlled conditions to the filter system is an essential step to mimic the practical applicability. Estimating the actual bacterial growth in cfu/cm2 under a longer incubation time is also necessary.
Hence, the research objectives and the experimental strategies are not appropriately planned. The scientific and practical purposes of this research are not clarified. The interpretation of the results are also not properly explored/discussed too. Overall, the manuscript is poorly written, not acceptable as per scientific standards of this field without having clear evidence of novelty and deep science study.
Author Response

(The authors gave the same response as above.)

Reviewer 3 Report
The authors describe a method of forming keratin/PA6 composites doped with Ag for use in filtration. The work appears interesting but it is recommended that the following issues be addressed before the manuscript is considered for publication.
The language needs to be improved as parts of the manuscript are difficult to understand.
Please cite the following values (means and standard deviation) in a table: mean fiber diameter, filter pore sizes, sizes of the bacteriostatic circle, mass of residues from TGA analyses, peak degradation temperatures, and peak melting temperatures.
Please indicate standard deviation by error bars in figures.
Some aspects are unclear.
- What is PM2.5 in the filtration QF tests?
- Why is it a benefit if the mass of thermal degradation residues increase, as appears to be suggested in line 218, p7.
- The evidence of lack of microbial growth in Fig 8 is unclear.
Author Response

(The authors gave the same response as above.)

Reviewer 4 Report
The content of this paper was to introduce the reparation of Ag doped keratin/PA6 air filtration membrane with enhanced antimicrobial properties. It provided useful information. However, some problems need to be considered.
The cited style of literature was not correct. Please check the Guide of authors
The unit should separate with words, for example 28.5 wt%, 3 h. Line 76-81, there were 25℃, 20%, 0.4,,l/h, 30kv and 625px. Please check all units in this paper.
The quality factor (QF) is an international standard, the standard of some societies related polymers, or just a person idea?
The same problem for WVT
Line 186-188, line 193-195, sat same thing as Figure 4.
In figure 8, that is only a visual method to prove the antibacterial activity of composite nanofibers. Is there other quantitative method?
Author Response

(The authors gave the same response as above.)

Round 2
Reviewer 1 Report
Overall the manuscript has improved a lot. Author are claiming that antibacterial property is greater for E. coli than S. aures. In general, It is opposite the former class is more resistant(Gram negative bacteria due to their cell wall) than the later(S. aures gram positive). Either provide more evidence if this is indeed true by CFU (colony forming unit) counts.
L53-56; L81-86 Please remove bold.
L410 Known? Lot of bacterium? both are gramatically incorrect.
Earlier Figure 7 is still there without ligand. please delet it.
Please write how authors has calculated diameter in nm range from SEM image.
Author Response
Dear Reviewers
Thank you for reading our manuscript and reviewing it again. We have sent the revised manuscript, and a version containing all the changes to be visible.
At the following, the point mentioned by the reviewers will be discussed:
Overall the manuscript has improved a lot. Author are claiming that antibacterial property is greater for E. coli than S. aures. In general, It is opposite the former class is more resistant(Gram negative bacteria due to their cell wall) than the later(S. aures gram positive). Either provide more evidence if this is indeed true by CFU (colony forming unit) counts
Answer:
Firstly, According to the previous literatures, some nanofibers content AgNPs may have a higher bacterial inhibition rate against E.coli than S.aureus (Food Packaging and Shelf Life 2018, 16, 129-137.; Applied Surface Science 2016, 387, 828-838.; Carbohydrate Polymers 2017, 165, 304-312). The composite membrane could achieve higher bacterial inhibition rate when the content of AgNPs was higher.
Secondly, the test method of durable antibacterial activity we previously used may be unreasonable and lead to errors in antibacterial results. Therefore We have modified the experimental scheme (iScience 2019, 19, 214-223) to make it more close to the practical application. We simulated the working environment of air filtration membrane and tested the bacterial inhibition rate of the composite membrane (Ag.10%-Ker.30%/PA6) before and after continuous operation for 120 h under the air filtration condition (PM2.5 > 1500 μg/m3) to evaluate the durable antibacterial activity of Ag.10%-Ker.30%/PA6 nanofibers membrane. The nanofibers membrane of Ag.10%-Ker.30%/PA6 maintained a high level of antibacterial effectiveness and the bacterial inhibition rate against S. aureus (decreased from 99.62% to 96.7%) and E. coli (decreased from 99.10% to 95.4%), which are shown at L348-355 P10 in revised manuscript.
L53-56; L81-86 Please remove bold.
Answer: We have deleted it, which are shown at L44-46 L63-65 P2 in revised manuscript.
L410 Known? Lot of bacterium? both are gramatically incorrect.
Answer: We have corrected it as suggested at L328 P10 in revised manuscript.
Earlier Figure 7 is still there without ligand. please delet it.
Answer: We have deleted it as suggested in revised manuscript.
Please write how authors has calculated diameter in nm range from SEM image.
Answer: The averaged diameter of nanofibers was measured from at least 100 nanofibers per sample by image-pro plus software, which is shown at L136 P4 in revised manuscript.
Reviewer 2 Report
The authors have thoughtfully and appropriately addressed the queries. Hence, I would recommend accepting the manuscript for publication.
Author Response
Dear Reviewers
Thank you for reading our manuscript again give it positive comments. We have sent the revised manuscript according to other reviewers. Thank you.
Reviewer 3 Report
Please explain all abbreviations. Fr example, the meaning of "TFE" in Line 70 on Page 2 is not clear.
Author Response
Dear Reviewers
Thank you for reading our manuscript and reviewing it again. We have sent the revised manuscript, and a version containing all the changes to be visible.
At the following, the point mentioned by the reviewers will be discussed:
Please explain all abbreviations. Fr example, the meaning of "TFE" in Line 70 on Page 2 is not clear.
Answer: HAADF-STEM is high-angle annular dark field-scanning transmission electron microscopy, which is shown at L21 P1 in revised manuscript.
PCL, TFE and 3T3 cells are Poly (ε‐caprolactone), Trifluoroethanol and a mouse fibroblast cell line, respectively, which are shown at L71-73 P2 in revised manuscript.
SDS is Sodium dodecyl sulfate, which is shown at L95 P3 in revised manuscript.
PBS buffer is phosphate buffer saline, which is shown at L187 P5 in revised manuscript.
PEO and PVA are polyethylene oxide and polyvinyl alcohol, which are shown at L237-238 P7 in revised manuscript.
EDS is energy dispersive spectrometer, which is shown at L240 P6 in revised manuscript.
Round 3
Reviewer 1 Report
I have gone through one of the MS you mentioned Carbohydrate Polymers 2017, 165, 304-312. In their abstract/discussion they only claimed that their polymer has antibacterial activity against both gram positive and negative bacteria without comparing which one is better. Their data also indicating some error. This must be an error because there trend is different except for 15min in E.coli. Reviewer of this MS failed to recognise this or may not have paid attention.
5 min 10 min 15 min
E.coli 1.42 1.77 3.04 - always lower than staph except for the 15min.
Staph 1.63 2.04 2.53
However, other two bacteria of the same groups are in right direction.
Please trust your hands when working on bench and also if you are arguing about something make sure you know the reason behind it. As I earlier mentioned that Gram negative are more resistance than positives because of different cell wall composition and this should not change regardless of concentration/time of incubation of antibacterial agent.
Now with the changed conditions the results are believable (even if it is small difference). You may want to provide BFE for both E. coli and Staph in abstract, conclusion where it is 96.8% for staph or can remove this line. are the numbers 15% and 18% for Staph and E. coli still the same? Everything else looks good except a minor change in 2.3 should have space before characteristics and 2.3.1 Line 129-149 has different line spacing than rest of the MS. I strongly recommend please go over the manuscript again before final submission to make sure there is no format or typo error exists.
Author Response
Dear Reviewers
Thank you for reading our manuscript and reviewing it, which will help us improve it to a better scientific level. We revised our manuscript, and some changes have taken place. So we have sent the revised manuscript, and a version containing all the changes to be visible.
At the following, the point mentioned by the reviewers will be discussed:
Reviewer 1:
I have gone through one of the MS you mentioned Carbohydrate Polymers 2017, 165, 304-312. In their abstract/discussion they only claimed that their polymer has antibacterial activity against both gram positive and negative bacteria without comparing which one is better. Their data also indicating some error. This must be an error because there trend is different except for 15min in E.coli. Reviewer of this MS failed to recognise this or may not have paid attention.
5 min 10 min 15 min
coli 1.42 1.77 3.04 - always lower than staph except for the 15min.
Staph 1.63 2.04 2.53
However, other two bacteria of the same groups are in right direction
Please trust your hands when working on bench and also if you are arguing about something make sure you know the reason behind it. As I earlier mentioned that Gram negative are more resistance than positives because of different cell wall composition and this should not change regardless of concentration/time of incubation of antibacterial agent.
Now with the changed conditions the results are believable (even if it is small difference). You may want to provide BFE for both E. coli and Staph in abstract, conclusion where it is 96.8% for staph or can remove this line. are the numbers 15% and 18% for Staph and E. coli still the same?
Answer: Firstly, we have added bacterial filtration efficiency (BFE) of Ag.10%-Ker.30%/PA6 composite membrane against E. coli using ASTM F2101-14 Standard Test Method. In this process, the number of E. coli in liquid culture medium was determined by colony count method, and the nutrient broth was diluted to the required concentration by PBS buffer. Then the liquid was used to prepare microbial aerosol by a spray suspension method and the challenge suspension was delivered to the nebulizer in Handersen pipeline for 1 min. The sample port with the nanofibers mats and not mats were counted with Anderson 6 level microbial aerosol sampler at a flow rate of 28.3 L/m. Then the specimen plates were incubated at 37℃ and colony count after 48 h. Bacterial Filtration efficiency of composite membrane against E. coli was 95.6%, which is shown at L349 P10 in revised manuscript.
Secondly, thank you very much for your theoretical guidance on bacterial resistance, which will be of great help to our future scientific research work. We have reviewed lots of previous literatures and found that the E. coli does have a higher resistance (PANA, 2018, 42, 10786-10791. Materials Science & Engineering C, 2019, 102, 616-622), which is consistent with the viewpoint you mentioned. However, the sterilization mechanism of nano-silver is not unique. The mechanism of the antibacterial activity of AgNPs is complex: (1) AgNps is highly active and these silver ions are released into the bacteria as a reservoir of silver ions, interacting with the thiol groups of enzymes and proteins producing the inhibition of their functions (Nanotechnology, 2007, 18, 1-10); (2) The silver nanoparticles can modulate signal transduction between bacteria stopping their growth (Arabian Journal of Chemistry, 2019, 12, 825-834); (3) Some authors have proposed that the formation of free radicals by nanoparticles can damage the cell membrane and make it porous. This also causes an electrochemical imbalance in the cells and allows the silver ions to pass through the plasma membrane into the cytoplasm of the bacterial cell and interact with the intracellular components resulting in permanent cell damage (Nanomaterials, 2018, 8, 1009). We're not sure which one plays a major role in our experiment. So we have redid a lot of bacterial inhibition rate (BI) tests on the Ag.1%-Ker.30%/PA6 composite membrane against S. aureus and E. coli (table 1), and the average inhibition rate against S. aureus and E. coli represents final bacterial inhibition rate, which are 20.5% and 19.9%, respectively. Table 1 shows two phenomenon which are the bacterial inhibition rate of E. coli higher than S. aureus and the bacterial inhibition rate of E. coli lower than S. aureus, respectively. It was not a one-sided phenomenon that the inhibition rate of E. coli is always lower than that of S. aureus. The results are shown at L364 P11 in revised manuscript.
Thanks again for your patient theoretical guidance, which makes the structure of this paper more rigorous and also improves our theoretical knowledge of bacterial resistance.
Table 1. The inhibition rate of S. aureus and E. coli in twenty repeated trials
|
|
1 |
2 |
3 |
4 |
5 |
6 |
7 |
8 |
9 |
10 |
11 |
|
E. coli |
19.7% |
20.8% |
18.9% |
21.3% |
19.3% |
17.6% |
18.7% |
20.4% |
20.6% |
22.3% |
20.8% |
|
S. aureus |
20.3% |
20.5% |
21.6% |
21.2% |
21.6% |
20.3% |
19.8% |
18.9% |
19.9% |
21.4% |
21.3% |
|
|
12 |
13 |
14 |
15 |
16 |
17 |
18 |
19 |
20 |
Average |
|
|
E. coli |
21.0% |
19.9% |
19.6% |
20.3% |
20.4% |
18.6% |
19.4% |
18.8% |
20.5% |
19.95% |
|
|
S. aureus |
19.7% |
20.6% |
21.3% |
18.9% |
22.6% |
20.3% |
21.2% |
19.8% |
19.9% |
20.65% |
|
Everything else looks good except a minor change in 2.3 should have space before characteristics and 2.3.1 Line 129-149 has different line spacing than rest of the MS.
Answer: We have corrected it as suggested, which are shown at L129-149 P4 in revised manuscript.
I strongly recommend please go over the manuscript again before final submission to make sure there is no format or typo error exists.
Answer: Thank you for your suggestion. We have checked the manuscript again.
